# Efficient Beam Tree Recursion

**Jishnu Ray Chowdhury**    **Cornelia Caragea**
Computer Science
University of Illinois Chicago
jraych2@uic.edu        cornelia@uic.edu

## Abstract

Beam Tree Recursive Neural Network (BT-RvNN) was recently proposed as an extension of Gumbel Tree RvNN and it was shown to achieve state-of-the-art length generalization performance in ListOps while maintaining comparable performance on other tasks. However, although better than previous approaches in terms of memory usage, BT-RvNN can be still exorbitantly expensive. In this paper, we identify the main bottleneck in BT-RvNN's memory usage to be the entanglement of the scorer function and the recursive cell function. We propose strategies to remove this bottleneck and further simplify its memory usage. Overall, our strategies not only reduce the memory usage of BT-RvNN by $10 - 16$ times but also create a new state-of-the-art in ListOps while maintaining similar performance in other tasks. In addition, we also propose a strategy to utilize the induced latent-tree node representations produced by BT-RvNN to turn BT-RvNN from a sentence encoder of the form $f : \mathbb{R}^{n \times d} \to \mathbb{R}^d$ into a token contextualizer of the form $f : \mathbb{R}^{n \times d} \to \mathbb{R}^{n \times d}$. Thus, our proposals not only open up a path for further scalability of RvNNs but also standardize a way to use BT-RvNNs as another building block in the deep learning toolkit that can be easily stacked or interfaced with other popular models such as Transformers and Structured State Space models. Our code is available at the link: https://github.com/JRC1995/BeamRecursionFamily.

## 1 Introduction

Recursive Neural Networks (RvNNs) [63] in their most general form can be thought of as a repeated application of some arbitrary neural function (the recursive cell) combined with some arbitrary halting criterion. The halting criterion can be dynamic (dependent on input) or static (independent of the input). From this viewpoint, nearly any neural network encoder in the deep learning family can be seen as a special instance of an RvNN. For example, Universal Transformers [13] repeat a Transformer [85] layer block as a recursive cell and adaptively halt by tracking the halting probability in each layer using some neural function [25, 3]. Deep Equilibrium Models (DEQ) [2] implicitly "repeat" a recursive cell function using some root-finding method which is equivalent to using the convergence of hidden states dynamics as the halting criterion. As Bai et al. [2] also showed, any traditional Transformer - i.e. stacked layer blocks of Transformers with non-shared weights can be also equivalently reformulated as a recursive repetition of a big sparse Transformer block with the halting criterion being some preset static upperbound (some layer depth set as a hyperparameter).

A broad class of RvNNs can also be viewed as a repeated application of a Graph Neural Network (GNN) [68, 91] - allowing iterative message passing for some arbitrary depth (determined by the halting criterion). Transformer layers can be seen as implementing a fully connected (all-to-all) graph with sequence tokens as nodes and attention-based edge weights. In natural language processing, we often encounter the use of stacks of GNNs (weight shared or not) to encourage message-passing through underlying linguistic structures such as dependency parses, constituency parses, discourse structures, abstract meaning representations, and the like [80, 39, 47, 59, 11, 90, 94].

37th Conference on Neural Information Processing Systems (NeurIPS 2023).

In many cases, such models are implemented with some static fixed number of layers or iterations. However, learning certain tasks in a length-generalizing fashion without prior knowledge of the sequence length distribution should require a dynamic halting function. For example, consider a simple arithmetic task: $7 \times (8+3)$. It is first necessary to process $8+3$ before doing the multiplication in the next step. A task like this can require structure-sensitive sequential processing where the sequential steps depend on the input length and nested operators. Although the above example is simple enough, we can have examples with arbitrary nested operators within arbitrarily complex parenthetical structures. Datasets like ListOps [57] and logical inference [7], among others, serve as a sanity check for any model's ability to perform similar structure-sensitive tasks. Natural language tasks, on the other hand, can sometimes hide systematic failures because of the distributional dominance of confounding statistical factors [54]. While, in some algorithmic tasks, Universal Transformer [13] can perform better than vanilla Transformers, both can still struggle in length generalization in logical inference or ListOps [71, 84]. This brings us to explore another alternative family of models—Tree-RvNNs with a stronger inductive bias than Universal Transformer but still possessing a form of dynamic halt that we discuss below.

**Tree Recursive Neural Networks:** Tree-RvNNs [79, 63, 77] (more particularly, constituency Tree RvNNs) treat an input sequence of vectors as the terminal nodes of some underlying latent tree. From this perspective, Tree-RvNNs sequentially build up the tree nodes in a bottom-up manner. Each node (terminal or non-terminal) in the tree will represent some span within the sequence (for text sequences, these would be words, phrases, clauses, and the like). The root node will represent the whole input sequence and thus can be used as an encoded representation of the whole input (for sentences, it can be treated as a sentence encoding). The nodes are generally in the form of vectors.

The non-terminal nodes are built bottom-up through the repeated use of some recursive cell composition function, say $rec$, starting from height $1$ (height $0$ being the terminal nodes) all the way up to the root. When building the representation of some node $p$ at height $t$, the Tree-RvNN uses the $rec$ function to take all the immediate child nodes of $p$ (from the previous heights $0$ to $t-1$) as input arguments and outputs the representation of $p$. Typically, we only consider binarized trees so the number of children as arguments will be a constant (two) for $rec$. Thus, it can be represented in the form $rec : \mathbb{R}^d \times \mathbb{R}^d \to \mathbb{R}^d$. Generally, we also constrain our consideration to only projective tree structures.[1] This implies that the model enforces only temporally contiguous node pairs be considered as candidate siblings for any parent node.[2] Finally, when the root representation is built the computation terminates. Therefore, reaching the root is the halting criterion. Since the tree structure is dependent on the input and the halting is dependent on the tree structure, the halting mechanism here is dynamic and thus can adapt with input complexity.

This approach has the potential to model mereological (part-whole) structures at different hierarchical scales in a systematic and dynamic manner.[3] As an example, let us consider that we have an input such as $(4 \times 3) + (7 + (8 \times 9))$. Then, in the ideal case, the Tree-RvNN order of operation would potentially yield: $rec(rec(rec(rec(4, \times)3), +), rec(rec(7, +), rec(rec(8, \times), 9)))$. As we can see, Tree-RvNNs have the potential to model sensitive order of operations considering input hierarchies which can be botched by unconstrained attention without careful inductive biases [12, 10, 29] or extensive pre-training[4]. We present an extended related work review in Appendix D.

**Motivation for Tree-RvNNs:** Although, at first glance Tree-RvNNs as formalized above may appear to have too strong of an inductive bias, there are a few reasons to be motivated to empirically explore the vicinity of these models:

1. Familiar Recurrent Neural Networks (RNNs) [19, 33] are special cases of Tree-RvNNs that follow an enforced chain-structured tree. So at the very least Tree-RvNNs have more flexibility than RNNs which have served as a fairly strong general model in the pre-Transformer era; and even now, optimized linear variants of RNNs [53] are making a comeback out-competing Transformers in long range datasets [26, 27, 60].

---

[1]This is done mainly to simplify the search space but assuming projective structures is also standard fare in Natural Language Processing (NLP) [41, 96].

[2]"temporal" refers to the original sequential order. The original order is preserved in every iteration.

[3]Similar inductive biases can be favorable in computer vision as well [76, 30, 74]

[4]Overall, Tree-RvNNs, as described above, can be also considered as a special case of DAG-GNN [83] where similar principles can apply. Exploring bottom-up tree node building through a more flexible space of DAG-structures can be a direction to consider in the future.

2. Often Tree-RvNNs are the core modules behind models that have been shown to productively length-generalize or compositionally generalize [71, 10, 46, 32, 5, 45, 52] in settings where other family of models struggle (at least without extensive pre-training).

3. The recursive cell function of Tree-RvNNs can still allow them to go outside their explicit structural constraints by effectively organizing information in their hidden states if necessary. The projective tree-structure provides only a rough contour for information flow through Tree-RvNNs which Tree-RvNNs can learn to go around just as an RNN can [7]. So in practice some of these constraints can be less concerning than one may think.

**Issues:** Unfortunately, the flexibility of Tree-RvNNs over RNNs comes at a cost. While in RNNs we could just follow on the same chain-structure tree (left-to-right or right-to-left) for any input, in Tree-RvNNs we have to also consider some way to dynamically induce a tree-structure to follow. This can be done externally—that is, we can rely on human inputs or some external parser. However, neither of them is always ideal. Here, we are focusing on *complete Tree-RvNN* setups that do their own parsing without any ground-truth tree information anywhere in the pipeline. Going this direction makes the implementation of Tree-RvNNs more challenging because it is hard to induce discrete tree structures through backpropagation. Several methods have been proposed to either induce discrete structures [31, 9, 66, 61], or approximate them through some continuous mechanism [10, 95, 72, 71]. Nevertheless, all these methods have their trade offs - some do not actually work in structured-sensitive tasks [57, 9, 72], others need to resort to reinforcement learning and an array of optimizing techniques [31] or instead use highly sophisticated architectures that can become practically too expensive in space or time or both [51, 71, 10]. Moreover, many of the above models [10, 31, 9] have been framed and used mainly as a sentence encoder of the form $f : \mathbb{R}^{n \times d} \to \mathbb{R}^d$ ($n$ being the sequence size). This formulation as sentence encoder limits their applicability as a deep learning building block - for example, we cannot use them as an intermediate block and send their output to another module of their kind or some other kind like Transformers because the output of a sentence encoder will just be a single vector.

**Our Contributions:** Believing that simplicity is a virtue, we direct our attention to Gumbel-Tree (GT) RvNN [9] which uses a simple easy-first parsing technique [23] to automatically greedily parse tree structures and compose sentence representations according to them (we provide a more extended discussion on related works in Appendix). Despite its simplicity, GT-RvNN relies on Straight-Through Estimation (STE) [4] which induces biased gradient, and has been shown empirically to fail in structure-sensitive tasks [57]. Yet, a recent approach - Beam Tree RvNN (BT-RvNN) [66] - promisingly shows that simply using beam search instead of the greedy approach succeeds quite well in structure-sensitive tasks like ListOps [57] and Logical Inference [7] without needing STE. Nevertheless, BT-RvNN can still have exhorbitant memory usage. Furthermore, so far it has been only tested as a sentence encoder. We take a step towards addressing these issues in this paper:

1. We identify a critical memory bottleneck in both Gumbel-Tree RvNN and Beam-Tree RvNN and propose strategies to fix this bottleneck (see §3). Our strategies reduce the peak memory usage of Beam-Tree RvNNs by 10-16 times in certain stress tests (see Table 4).

2. We propose a strategy to utilize the intermediate tree nodes (the span representations) to provide top-down signals to the original terminal representations using a parent attention mechanism. This allows a way to contextualize token representations using RvNNs enabling us to go beyond sentence-encoding (see §4).

3. We show that the proposed efficient variant of BT-RvNN incurs marginal accuracy loss if at all compared to the original—and in some cases even outperforms the original by a large margin (in ListOps).

## 2   Existing Framework

Here we first describe the existing framework used in Choi et al. [9] and Ray Chowdhury and Caragea [66]. In the next section, we identify its weakness in terms of memory consumption and resolve it.

**Task Structure:** As in related prior works [9, 10, 71], we start with our focus (although we will expand - see §4) on exploring the use of RvNNs as sentence encoders of the form: $f : \mathbb{R}^{n \times d} \to \mathbb{R}^d$. Given a sequence of $n$ vectors of size $d$ as input (of the form $\mathbb{R}^{n \times d}$), $f$ compresses it into a single vector (of the form $\mathbb{R}^d$). Sentence encoders can be used for sentence-pair comparison or classification.

**Components:** The core components of this framework are: a scoring function $scorer : \mathbb{R}^d \rightarrow \mathbb{R}$ and a recursive cell $rec : \mathbb{R}^d \times \mathbb{R}^d \rightarrow \mathbb{R}^d$. The $scorer$ is implemented as $scorer(v) = W_v v$ where $W_v \in \mathbb{R}^{1 \times d}$. The $rec(child_l, child_r)$ function is implemented as below:

$$\begin{bmatrix} l, \\ r, \\ g, \\ h \end{bmatrix} = \text{GeLU}\left(\begin{bmatrix} child_l; \\ child_r \end{bmatrix} W_1^{rec} + b_1\right) W_2^{rec} + b_2 \tag{1}$$

$$p = LN(\sigma(l) \odot child_l + \sigma(r) \odot child_r + \sigma(g) \odot h) \tag{2}$$

Here - $\sigma$ is $sigmoid$; $[;]$ represents concatenation; $p$ is the parent node representation built from the children $child_l$ and $child_r$, $W_1^{rec} \in \mathbb{R}^{2 \cdot d \times d_{cell}}$; $b_1 \in \mathbb{R}^{d_{cell}}$; $W_2^{rec} \in \mathbb{R}^{d_{cell} \times 4 \cdot d}$; $b_2 \in \mathbb{R}^d$, and $l, r, g, h \in \mathbb{R}^d$. $LN$ is layer normalization. $d_{cell}$ is generally set as $4 \times d$. Overall, this is the Gated Recursive Cell (GRC) that was originally introduced by Shen et al. [71] and has been consistently shown to be superior [71, 10, 66] to earlier RNN cells like Long Short Term Memory Networks (LSTM) [33, 80].

Note that these models generally also apply an initial transformation layer to the terminal nodes before starting up the RvNN. Similar to [71, 10, 66], we apply a single linear transform followed by a layer normalization as the initial transformation.

**Greedy Search Tree Recursive Neural Networks:** Here we describe the implementation of Gumbel-Tree RvNN [9]. Assume that we have a sequence of hidden states of the form $(h_1^t, h_2^t, ...h_n^t)$ in some intermediate recursion layer $t$. For the recursive step in that layer, first all possible parent node candidates are computed as:

$$p_1^t = rec(h_1^t, h_2^t), p_2^t = rec(h_2^t, h_3^t), \ldots, p_{n-1}^t = rec(h_{n-1}^t, h_n^t) \tag{3}$$

Second, each parent candidate node $p_i^t$ is scored as $e_i^t = scorer(p_i^t)$. Next, the index of the top score is selected as $j = argmax(e_{1:n-1}^t)$. Finally, now the update rule for the next recursion can be expressed as:

$$h_i^{t+1} = \begin{cases} h_i^t & i < j \\ rec(h_i^t, h_{i+1}^t) & i = j \\ h_{i+1}^t & i > j \end{cases} \tag{4}$$

Note that this is essentially a greedy tree search process where in each turn all the locally available choices (parent candidates) are scored and the maximum scoring choice is selected. In each iteration, the sequence size is decreased by 1. In the final step only one representation will be remaining (the root node). At this point, however, the tree parsing procedure is not differentiable because it purely relies on an argmax. In practice, reinforcement learning [31], STE with gumbel softmax [9], or techniques like SPIGOT [61] have been used to replace argmax. Below we discuss another alternative and our main focus.

**Beam Search Tree Recursive Neural Networks (BT-RvNN):** Seeing the above algorithm as a greedy process provides a natural extension through beam search as done in Ray Chowdhury and Caragea [66]. BT-RvNN replaces the argmax in Gumbel-Tree RvNN with a stochastic Top-$K$ [42] that stochastically extracts both the $K$ highest scoring parent candidates and the $K$ corresponding log-softmaxed scores. The process collects all parent node compositions and also accumulates (by addition) log-softmaxed scores for each selected choices in corresponding beams. With this, the end result is $K$ beams of accumulated scores and $K$ beams of root node representations. The final representation is a softmax-based marginalization of the $K$ root nodes: $\sum_i \frac{\exp(s_i)}{\sum_j \exp(s_j)} \cdot b_i$ where $b_i$ is the root node representation of beam $i$ and $s_i$ is the accumulated (added) log-softmaxed scores at every iteration for beam $i$. Doing this enabled BT-RvNN to improve greatly [66] over greedy Gumbel-Tree recursion [9]. However, BT-RvNN can also substantially increase the memory usage, which makes it inconvenient to use.

## 3 Bottleneck and Solution

In this section, we identify a major bottleneck in the memory usage that exists in the above framework (both for greedy-search and beam-search) that can be adjusted for.

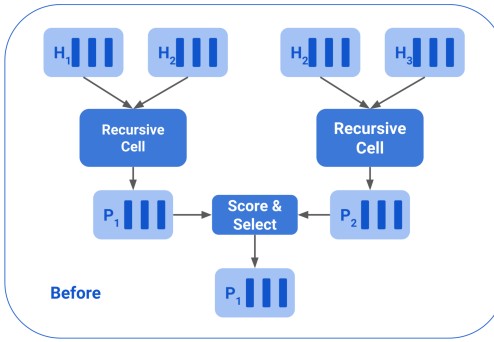 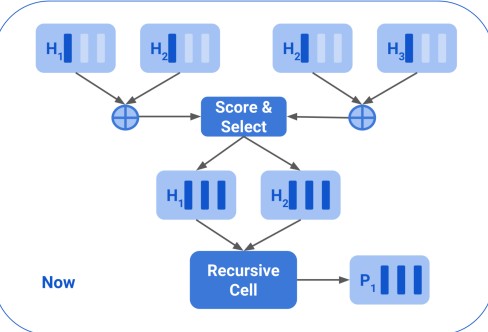

Figure 1: Visualization of the contrast between the existing framework (left) and the proposed one (right). $H_1, H_2, H_3$ are the input representations in the iteration. The possible contiguous pairs of them are candidate child pairs for nodes to be built in this iteration. On the left side, we see each pair is in parallel fed to the recursive cells to create their corresponding candidate parent representations. Then they are scored and one parent ($P_1$) is selected. On the right side (our approach), each child pair candidate is directly scored. The faded colored bars in $H_1, H_2, H_3$ represent sliced away vector values. The scoring function then selects one child pair. Then only that specific selected child pair is composed using the recursive cell to create the parent representation ($P_1$) not wasting unnecessary compute by applying the recursive cell for other non-selected child pairs.

**Bottleneck:** The main bottleneck in the above existing framework is Eqn. 3. That is, the framework runs a rather heavy recursive cell (concretely, the GRC function in Eqn. 2) **in parallel** for every item in the sequence and **for every iteration**. In contrast, RNNs could use the same GRC function but for **one position at a time** sequentially - taking very little memory. While Transformers also use similarly big feedforward networks like GRC in parallel for all hidden states - they have fixed a number of layers - whereas BT-RvNNs may have to recurse for hundreds of layers depending on the input making this bottleneck more worrisome. However, we think this bottleneck can be highly mitigated. Below, we present our proposals to fix this bottleneck.

### 3.1 Efficient Beam Tree Recursive Neural Network (EBT-RvNN)

Here, we describe our new model EBT-RvNN. It extends BT-RvNN by incorporating the fixes below. We present the contrast between the previous method and our current method visually in Figure 1.

**Fix 1 (new scorer):** At any iteration $t$, we start only with some sequence $(h_1^t, \ldots, h_n^t)$. In the existing framework, starting from this the function to compute the score for any child-pair will look like:

$$e_i = scorer \circ rec(h_i, h_{i+1}) \tag{5}$$

This is, in fact, the only reason for which we need to apply $rec$ to all positions (in Eqn. 3) at this iteration because that is the only way to get the corresponding score; that is, currently the score computation entangles the recursive cell $rec$ and the $scorer$. However, there is no clear reason to do this. Instead we can just replace $scorer \circ rec$ (in Eqn. 5) with a single new scorer function ($scorer_{new} : \mathbb{R}^{2 \times d} \to \mathbb{R}$) that directly interacts with the concatenation of $(h_i, h_{i+1})$ without the $rec$ as an intermediate step - and thus disentangling it from the scorer. We use a parameter-light 2-layered MLP to replace $scorer \circ rec$:

$$e_i = scorer_{new}(h_i, h_{i+1}) = \text{GeLU}([h_i; h_{i+1}]W_1^s + b_1^s)W_2^s + b_2^s \tag{6}$$

Here, $W_1^s \in \mathbb{R}^{2 \cdot d \times d_s}, W_1^s 2 \in \mathbb{R}^{d_s \times 1}, b_1^s \in \mathbb{R}^{d_s}, b_2^s \in \mathbb{R}$. Since lightness of the $scorer_{new}$ function is critical for lower memory usage (this has to be run in parallel for all contiguous pairs) we set $d_s$ to be small ($d_s = 64$). In this formulation, the $rec$ function will be called only for the selected contiguous pairs (siblings) in Eqn. 4.

**Fix 2 (slicing):** While we already took a major step above in making BT-RvNN more efficient, we can still go further. It is unclear whether the full hidden state vector size $d$ is necessary for parsing decisions. Parsing typically hangs on more coarse-grained abstract information - for example, when doing arithmetic while precise numerical information needs to be stored in the hidden states for future

computation, the exact numerical information is not relevant for parsing - only the class of being a number should suffice. Thus, we assume that we can project the inputs into a low-dimensional space for scoring. One way to do that is to use a linear layer. However, parallel matrix multiplications on the full hidden state can be still costly when done for each hidden state in the sequence in every recursion. So, instead, we can allow the initial transformation or the RvNN itself to implicitly learn to organize parsing-related information in some sub-region of the vector. We can treat only the first $min(d_s, d)$ (out of $d$) as relevant for parsing decisions. Then we can simply slice the first $min(d_s, d)$ out before sending the candidate child pairs to the scoring function. Thus, the score computation can be presented as below:

$$e_i = scorer_{new}(h_i[0 : min(d_s, d)], h_{i+1}[0 : min(d_s, d)]) \tag{7}$$

So, now $W_1^s \in I\!R^{2 \cdot min(d_s, d)} \times d_s$. As before we keep $d_s$ small ($d_s = 64$). Now, the total hidden state size ($d$) can be kept as high as needed to preserve overall representation capacity without making the computation scale as much with increasing $d$. The model can now just learn through gradient descent to organize parsing relevant features in the first $min(d_s, d)$ values of the hidden states because only through them will gradient signals related to parsing scores propagate. Note that unlike [31], we are not running a different $rec$ function for the parsing decisions. The parsing decision in our case still depends on the output of the same single recursive cell but from the previous iterations (if any).

**No OneSoft:** Ray Chowdhury and Caragea [66] also introduced a form of soft top-$k$ function (OneSoft) for better gradient propagation in BT-RvNN. While we still use that as a baseline model, we do not include OneSoft in EBT-RvNN. This is because OneSoft generally doubles the memory usage and EBT-RvNN already runs well without it. The combination of EBT-RvNN with OneSoft can be studied more in the future, but it is a variable that we do not focus on in this study.

None of the fixes here makes any strict asymptotic difference in terms of sequence length but it does lift a large overhead that can be empirically demonstrated (see Table 1).

# 4 Beyond Sentence Encoding

As discussed before many of the previous models [9, 10, 31, 66] in this sphere that focus on competency on structure-sensitive tasks have been framed to work only as a sentence encoder of the form $f : I\!R^{n \times d} \rightarrow I\!R^d$. Taking a step further, we also explore a way to use bottom-up Tree-RvNNs for token-level contextualization, i.e., to make it serve as a function of the form $f : I\!R^{n \times d} \rightarrow I\!R^{n \times d}$. This allows Tree-RvNNs to be stackable with other deep learning modules like Transformers.

Here, we consider whether we can re-formalize EBT-RvNNs for token contextualization. In EBT-RvNN[5], strictly speaking, the output is not just the final sentence encoding (the root encoding), but also the intermediate non-terminal tree nodes. Previously, we ignored them after we got the root encoding. However, using them can be the key to creating a token contextualization out of EBT-RvNNs. Essentially, what EBT-RvNN will build is a tree structure with node representations - the terminal nodes being the initial token vectors, the root node being the overall sentence encoding vector, and the non-terminal nodes representing different scales of hierarchies as previously discussed.

Under this light, one way to create a token contextualization is to contextualize the terminal representations based on higher-level composite representations at different scales or hierarchies of which the terminal representation is a part of. In other words, while we use a bottom-up process of building wholes from parts during sentence encoding, for token contextualization, we can implement a top-down process of contextualizing parts from the wholes that it compose.

A similar idea is used by Teng et al. [82] to recursively contextualize child node representations based on their immediate parent node using another recursive cell starting from the root and ending up at the terminal node representations. The contextualized terminal node representations can then become the contextualized token representations. But this idea requires costly sequential operations.

An alternative - that we propose - is to allow the terminal nodes to attend [1] to the non-terminal nodes to retrieve relevant information to different scales of hierarchies. More precisely, if we want the terminal nodes to be contextualized by the wholes that they compose then we want to restrict the attention to only the parents (direct or indirect). This can be done by creating an attention mask based

---

[5]The principles discussed here also apply to them Gumbel Tree RvNNs, BT-RvNNs, and the like.

on the induced tree structures. In practice, we allow every terminal node as queries to attend to every node as keys and values but use a mask to allow attention only if the key represents the same node as that represented by the query or the key represents some parent (direct or indirect) of the node represented by the query. We also implement a relative positional encoding to bias attention [70] - using the difference in heights of the nodes as the relative distance. In essence, we are proposing the use of a form of graph attention network [86]. This attention mechanism can be repeated for iterative refinement of the token representations through multiple layers of message-passing.

In the case of EBT-RvNNs, we can create separate token representations for each beam and then marginalize them based on beam scores. We describe our concrete setup briefly below but more details are presented in Appendix F.

**Step 1:** First, we begin with beams of representations before they were marginalized. This allows us to access discrete edges connecting parents for every beam. As a setup, we have some $b$ beams of tree node representations and their structural information (edge connections).

**Step 2:** We use Gated Attention Unit (GAU) [36], a modern Transformer variant, as the attention mechanism block. We use the terminal node representations as queries ($Q$) and all the non-terminal nodes as keys ($K$) and values ($V$). We use GAU, like a graph attention network [86], by using an adjacency matrix $A$ as an attention mask. $A_{ij}$ is set as 1 if and only if $Q_i$ is a child of $K_j$ based on our tree extraction. Otherwise $A_{ij}$ is set as 0. Thus attention is only allowed from parents to their terminal children (direct or indirect).

**Step 3:** We implement a basic relative positional encoding - similar to that of Raffel et al. [64]. The only difference is that for us, the relative distances are the relative height distances.

**Step 4:** The GAU layer is repeated for iterative refinement of terminal node representations. We repeat for two iterations since this is an expensive step.

**Step 5:** As before, we marginalize the beams based on the accumulated log-softmax scores after the terminal node representations are contextualized.

**Use Case:** In theory, the contextualized terminal node representations that are built up in Step 5 can be used for any task like sequence labelling or masked language modeling. At this point, we explore one specific use case - sentence-pair matching tasks (natural language inference and paraphrase detection). For these tasks we have two input sequences that we need to compare. Previously we only created sentence encoding for each sequences and made the vectors interact, but now we can make the whole of two sequences of contextualized terminal-node embeddings interact with each other through a stack of GAU-based self-attention. This is an approach that we use for some of the sentence-matching tasks in Natural Language Processing (Table 3). The models are trained end to end. We discuss the technical details about these architectural setups more explicitly in Appendix F.

## 5 Experiments and Results

### 5.1 Model Nomenclature

**Sentence Encoder models:** **Transformer** refers to Transformers [85]; **UT** refers to Universal Transformers [13]; **CRvNN** refers to Continuous Recursive Neural Network [10]; **OM** refers to Ordered Memory [71]; **BT-GRC** refers to **BT-RvNN** implemented with GRC [66]; **BT-GRC OS** refers to BT-GRC combined with OneSoft (OS) Top-$K$ function [66]; **EBT-GRC** refers to our proposed EBT-RvNN model with GRC; **GT-GRC** refers to Gumbel-Tree RvNN [9] but with GRC as the recursive cell; **EGT-GRC** refers to GT-GRC plus the fixes that we propose.

**Sentence Interaction Models:** Sequence interaction models refer to the models in the style described in Section 4. These models use some deeper interaction between contextualized token representations from both sequences without bottlenecking the interactions through a pair of vectors. We use **EBT-GAU** to refer to the approach described in Section 4. **EGT-GAU** refers to a new baseline which uses the same framework as EBT-GAU except it replaces the Beam-Tree-Search with greedy STE gumbel-softmax based selection as in [9]. **GAU** refers to a plain stack of Gated Attention Units [36] (made to approximate the parameters of EBT-GAU) that do not use any Tree-RvNNs and is trained directly on <SEP> concatenated sequence pairs.

Table 1: Empirical time and (peak) memory consumption for various models on an RTX A6000. Ran on 100 ListOps data with batch size 1 and the same hyperparameters as used on ListOps on various sequence lengths. (-slice) indicates EBT-GRC without slicing from Fix 2, (512) indicates EBT-GRC with the hidden state dimension ($d$) set as 512 (instead of 128).(512,-slice) represents EBT-GRC with 512 dimensional hidden state size and without slicing.

| | Sequence Lengths | | | | | | | |
|---|---|---|---|---|---|---|---|---|
| **Model** | $200-250$ | | $500-600$ | | $900-1000$ | | $1500-2000$ | |
| | Time (min) | Memory (GB) | Time (min) | Memory (GB) | Time (min) | Memory (GB) | Time (min) | Memory (GB) |
| OM | 8.0 | 0.09 | 20.6 | 0.21 | 38.2 | 0.35 | 76.6 | 0.68 |
| CRvNN | 1.5 | 1.57 | 4.3 | 12.2 | 8.0 | 42.79 | OOM | OOM |
| GT-GRC | 0.5 | 0.35 | 2.1 | 1.95 | 3.5 | 5.45 | 7.1 | 21.76 |
| EGT-GRC | 1 | 0.07 | 2.5 | 0.3 | 4.3 | 0.81 | 8.5 | 3.15 |
| BT-GRC | 1.1 | 1.71 | 2.6 | 9.82 | 5.1 | 27.27 | OOM | OOM |
| BT-GRC OS | 1.4 | 2.74 | 4.0 | 15.5 | 7.1 | 42.95 | OOM | OOM |
| EBT-GRC | 1.2 | 0.19 | 3.2 | 1.01 | 5.5 | 2.78 | 10.5 | 10.97 |
| $-$ slice | 1.2 | 0.35 | 3.2 | 1.95 | 5.4 | 5.4 | 10.3 | 21.12 |
| (512) | 1.2 | 0.41 | 3.3 | 1.29 | 5.6 | 3.13 | 12.1 | 11.41 |
| $(512,-$ slice) | 1.2 | 1.55 | 3.3 | 7.77 | 5.5 | 21.02 | OOM | OOM |

## 5.2 Efficiency Analysis

In Table 1, we compare the empirical time-memory trade offs of the most relevant Tree-RvNN models (particularly those that are competent in ListOps and logical inference). We use CRvNN in the no halting mode as [66] because otherwise it can start to halt trivially because of limited training data. For the splits of lengths $200-1000$ we use the data shared by Havrylov et al. [31]; for the $1500-2000$ split we sample from the training set of LRA listops [81].

We can observe from the table that EBT-GRC achieves better memory efficiency among all the strong RvNN contenders (GT-GRC and EGT-GRC fail on ListOps/Logical inference) except for OM. However, OM's memory efficiency comes with a massive cost in time, being nearly 8 times slower than EBT-GRC. Compared to BT-GRC OS's 43GB peak memory consumption in 900-1000 sequence length from [66], the memory consumption of EBT-GRC is reduced to only 2.8GB. Even compared to BT-GRC, the reduction is near ten times. EBT-GRC even outperforms the original greedy GT-GRC used in Choi et al. [9]. Removing the slicing from the full model EBT-GRC (i.e., $-$slice) can substantially increase the memory cost. This becomes most apparent when training with higher hidden state size (compare (512) vs. (512,$-$slice)). This shows the effectiveness of slicing.

## 5.3 Results

Hyperparameters are presented in Appendix G, architecture details are presented in Appendix F, task details are provided in Appendix B and additional results (besides what is presented below) in logical inference and text classification are provided in Appendix C.

**List Operations (ListOps):** The task of ListOps consist of hierarchical nested operations that neural networks generally struggle to solve particularly in length-generalizable settings. There are only a few known contenders that achieve decent performance in the task [31, 10, 71, 66]. For this task we use the original training set [57] with the length generalization splits from Havrylov et al. [31], the argument generalization splits from Ray Chowdhury and Caragea [66], and the LRA test set from Tay et al. [81]. The different splits test the model in different out-of-distribution settings (one in unseen lengths, another in an unseen number of arguments, and another in both unseen lengths and arguments). Remarkably, as can be seen from Table 2, EBT-GRC outperforms most of the previous models in accuracy - only being slightly behind OM for some argument generalization splits. EBT-GRC $-$Slice represents the performance of EBT-GRC without slicing. It shows that slicing in fact improves the accuracy as well in this context but even without slicing the model is better than BT-GRC or BT-GRC OS.

Table 2: Accuracy on ListOps. For our models, we report the median of 3 runs. Our models were trained on lengths $\leq 100$, depth $\leq 20$, and arguments $\leq 5$. * represents results copied from [71]. We bold the best results that do not use gold trees. Subscript represents standard deviation. As an example, $90_1 = 90 \pm 0.1$

| Model | near-IID | Length Gen. | | | Argument Gen. | | LRA |
|---|---|---|---|---|---|---|---|
| (Lengths) | $\leq 1000$ | 200-300 | 500-600 | 900-1000 | 100-1000 | 100-1000 | 2000 |
| (Arguments) | $\leq 5$ | $\leq 5$ | $\leq 5$ | $\leq 5$ | 10 | 15 | 10 |
| *With gold trees* | | | | | | | |
| GoldTreeGRC | $99.9_{.2}$ | $99.9_{.9}$ | $99.8_1$ | $100_{.5}$ | $81.2_{28}$ | $79.5_{14}$ | $78.5_{29}$ |
| *Baselines without gold trees* | | | | | | | |
| Transformer * | $57.4_4$ | — | — | — | — | — | — |
| UT * | $71.5_{78}$ | — | — | — | — | — | — |
| GT-GRC | $75_{4.6}$ | $47.7_{8.4}$ | $42.7_{2.8}$ | $37.53_{37}$ | $50.9_{15}$ | $51.4_{16}$ | $45.3_{12}$ |
| EGT-GRC | $84.2_{19}$ | $51.3_{37}$ | $42.9_{35}$ | $34.4_{35}$ | $44.7_{17}$ | $40.8_{16}$ | $34.4_{14}$ |
| OM | $\mathbf{99.9_{.3}}$ | $99.6_7$ | $92.4_{13}$ | $76.3_{13}$ | $\mathbf{83.2_{24}}$ | $76.3_{38}$ | $\mathbf{79.3_{18}}$ |
| CRvNN | $99.7_{2.8}$ | $98.8_{11}$ | $97.2_{23}$ | $94.9_{49}$ | $66.6_{40}$ | $43.7_{38}$ | $55.38_{44}$ |
| BT-GRC | $99.4_{2.7}$ | $96.8_{10}$ | $93.6_{22}$ | $88.4_{27}$ | $75.2_{28}$ | $59.1_{79}$ | $63.4_{57}$ |
| BT-GRC OS | $99.6_{5.4}$ | $97.2_{35}$ | $94.8_{65}$ | $92.2_{86}$ | $73.3_{64}$ | $63.1_{92}$ | $66.1_{101}$ |
| EBT-GRC | $\mathbf{99.9_{0.3}}$ | $\mathbf{99.7_{2.4}}$ | $\mathbf{99.5_1}$ | $\mathbf{99.3_5}$ | $82.5_{13}$ | $\mathbf{79.6_{8.7}}$ | $\mathbf{79.3_{6.5}}$ |
| EBT-GRC − Slice | $99.7_3$ | $98.6_{12}$ | $98.4_{17}$ | $98.6_{14}$ | $79.3_{20}$ | $74.4_{37}$ | $75.5_{25}$ |

**Logical Inference and Sentiment Classification:** We show the results of our models in formal logical inference (another dataset where only RvNN-like models have shown some success) and sentiment classification in Appendix C. There we show that our EBT-GRC can easily keep up on these tasks with BT-GRC despite being much more efficient.

**NLI and Paraphrase Detection:** As we can see from Table 3, although EBT-GRC does not strictly outperform BT-GRC or BT-GRC OS, it remains in the same ballpark performance. The sentence interaction models, unsurprisingly, tend to have higher scores that sentence encoder modes because of their more parameters and more interaction space. We do not treat them commensurate here. Among the sequence interaction models, our EBT-GAU generally outperforms both baseline models in its vicinity - GAU and EGT-GAU. Even when used in conjunction with a Transformer, beam search still maintains some usefulness over simpler STE-based greedy search (EGT-GAU) and it shows some potential against pure Transformer stacks as well (GAU).

## 6 Conclusion

We identify a memory bottleneck in a popular RvNN framework [9] which has caused BT-RvNN [66] to require more memory than it needs to. Mitigating this bottleneck allows us to reduce the memory consumption of EBT-RvNN (an efficient variant of BT-RvNN) by 10-16 times without much other cost and while preserving similar task performances (and sometimes even beating the original BT-RvNN). The fixes also equally apply to any model using the framework including the original Gumbel Tree model [9]. We believe our work can serve as a basic baseline and a bridge for the development of more scalable models in the RvNN family and beyond.

## 7 Limitations

Although our proposal improves upon the computational trade-offs over some of the prior works [10, 9, 66, 71], it can be still more expensive than standard RNNs although we address this limitation, to an extent, in our concurrent work [65]. Moreover, our investigation of utilizing bottom-up Tree-RvNNs for top-down signal (without using expensive CYK models [15, 16]) is rather preliminary (efficiency being our main focus). This area of investigation needs to be focused more in the future. Moreover, although our proposal reduces memory usage it does not help much on accuracy scores compared with other competitive RvNNs.

Table 3: Mean accuracy and standard deviaton on SNLI [6], QQP, and MNLI [89]. Hard represents the SNLI test set from Gururangan et al. [28], Break represents the SNLI test set from Glockner et al. [22]. Count. represents the counterfactual test set from Kaushik et al. [38]. $\text{PAWS}_{QQP}$ and $\text{PAWS}_{WIKI}$ are adversarial test sets from Zhang et al. [97], ConjNLI is the dev set from Saha et al. [67], NegM,NegMM,LenM,LenMM are Negation Match, Negation Mismatch, Length Match, Length Mismatch stress test sets from Naik et al. [56] respectively. Our models were run 3 times on different seeds. Subscript represents standard deviation. As an example, $90_1 = 90 \pm 0.1$

| Models | SNLI Training | | | | QQP Training | | |
| --- | --- | --- | --- | --- | --- | --- | --- |
| | IID | Hard | Break | Count. | IID | $\text{PAWS}_{QQP}$ | $\text{PAWS}_{Wiki}$ |
| (Sequence Encoder Models) | | | | | | | |
| CRvNN | $85.3_2$ | $\mathbf{70.6_4}$ | $55.3_{17}$ | $59.8_6$ | $\mathbf{84.8_3}$ | $34.8_7$ | $46.6_6$ |
| OM | $\mathbf{85.5_2}$ | $70.6_3$ | $\mathbf{67.4_9}$ | $\mathbf{59.9_2}$ | $84.6_0$ | $\mathbf{38.1_7}$ | $45.6_8$ |
| BT-GRC | $84.9_1$ | $70_5$ | $51_{14}$ | $59_4$ | $84.7_5$ | $36.9_{17}$ | $46.4_{12}$ |
| BT-GRC OS | $84.9_1$ | $70.3_6$ | $53.29_{10}$ | $58.6_3$ | $84.2_2$ | $37.1_8$ | $46.3_6$ |
| EBT-GRC | $84.7_4$ | $69.9_8$ | $55.6_{20}$ | $58.1_1$ | $84.3_2$ | $36.9_5$ | $\mathbf{47.5_5}$ |
| (Sequence Interaction Models) | | | | | | | |
| GAU | $87_2$ | $74.2_4$ | $69.40_{34}$ | $66.4_{2.5}$ | $83.6_1$ | $38.6_{14}$ | $47.3_1$ |
| EGT-GAU | $87.1_0$ | $74.8_4$ | $66.1_{12}$ | $66.2_2$ | $83.5_4$ | $39.4_{31}$ | $\mathbf{49.3_{13}}$ |
| EBT-GAU | $\mathbf{87.5_2}$ | $\mathbf{75.7_3}$ | $\mathbf{70_{26}}$ | $\mathbf{67.6_4}$ | $\mathbf{83.9_2}$ | $\mathbf{42.3_{35}}$ | $47.2_8$ |

| Models | MNLI Training | | | | | | |
| --- | --- | --- | --- | --- | --- | --- | --- |
| | Match | MM | ConjNLI | NegM | NegMM | LenM | LenMM |
| (Sequence Encoder Models) | | | | | | | |
| CRvNN | $72.2_4$ | $72.6_5$ | $\mathbf{41.7_{10}}$ | $52.8_6$ | $53.8_{4.2}$ | $62_{44}$ | $63.3_{47}$ |
| OM | $\mathbf{72.5_3}$ | $\mathbf{73_2}$ | $\mathbf{41.7_4}$ | $50.9_7$ | $51.7_{13}$ | $56.5_{33}$ | $57.06_{31}$ |
| BT-GRC | $71.6_2$ | $72.3_1$ | $40.7_6$ | $\mathbf{53.7_{37}}$ | $\mathbf{54.8_{43}}$ | $64.7_6$ | $66.4_5$ |
| BT-GRC OS | $71.7_1$ | $71.9_2$ | $41.2_9$ | $53.2_2$ | $54.2_5$ | $\mathbf{65.6_13}$ | $\mathbf{66.7_9}$ |
| EBT-GRC | $72.1_2$ | $72_1$ | $40.93_0$ | $52.33_{23}$ | $53.28_{22}$ | $64.92_{10}$ | $66.4_{10}$ |
| (Sequence Interaction Models) | | | | | | | |
| GAU | $76.4_3$ | $76.5_2$ | $\mathbf{53.5_{12}}$ | $48.2_{11}$ | $48.24_{11}$ | $69.6_{20}$ | $70.6_{22}$ |
| EGT-GAU | $75.1_2$ | $75.5_3$ | $53.1_4$ | $48.7_{14}$ | $48.6_{14}$ | $69.8_{13}$ | $70.6_{11}$ |
| EBT-GAU | $\mathbf{76.5_1}$ | $\mathbf{76.9_2}$ | $53.3_{18}$ | $\mathbf{49.2_4}$ | $\mathbf{49_3}$ | $\mathbf{71.6_{18}}$ | $\mathbf{72.5_{19}}$ |

## 8   Acknowledgments

This research is supported in part by NSF CAREER award #1802358, NSF IIS award #2107518, and UIC Discovery Partners Institute (DPI) award. Any opinions, findings, and conclusions expressed here are those of the authors and do not necessarily reflect the views of NSF or DPI. We thank our anonymous reviewers for their constructive feedback.

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

# A   Appendix Organization

In Section B, we describe the settings of all the tasks and datasets that we have tested our models on. In Section C, we provide additional results on logical inference and sentiment classification. Then, in Section D, we present an extended survey of related works. In Section E, we present the pseudocode for EBT-RvNN. In Section F, we detail our architecture setup including the sequence-interaction models. In Section G, we provide our hyperparameters.

# B   Task Details

**ListOps:** ListOps was introduced by Nangia and Bowman [57] and is a task for solving nested lists of mathematical operations. It is a 10-way classification task. Similar to Chowdhury and Caragea [10], we train our models on the original training set with all samples $\geq 100$ sequence lengths filtered out. We use the original development set for validation. We test on the following sets: the original test set (near-IID split); the length generalization splits from Havrylov et al. [31] that include samples of much higher lengths; the argument generalization splits from Ray Chowdhury and Caragea [66] that involve an unseen number of maximum arguments for each operator; and the LRA split (which has both higher sequence length and higher argument number) from Tay et al. [81].

**Logical Inference:** Logical Inference was introduced by Bowman et al. [7] and is a task that involves classifying fine-grained inferential relations between two given sequences in a form similar to that of formal sentences of propositional logic. Similar to Tran et al. [84], our models were trained on splits with logical connectives $\leq 6$. We show the results in OOD test sets with logical connections 10-12. We use the same splits as Shen et al. [71], Tran et al. [84], Chowdhury and Caragea [10].

**SST5:** SST5 is a fine-grained 5-way sentiment classification task introduced by Socher et al. [79]. We use the original splits.

**IMDB:** IMDB is a binary sentiment classification task from Maas et al. [49]. We use the same train, validation, and IID test sets as created in Ray Chowdhury and Caragea [66]. We also use the contrast set Gardner et al. [21] and counterfactual set Kaushik et al. [38] as additional test splits.

**QQP:** QQP[6] [37] is a task of classifying whether two given sequences in a pair are paraphrases of each other or not. Following prior works Wang et al. [88], we randomly sample $10,000$ samples for validation and IID test set such that for each split $5,000$ samples are maintained to be paraphrases and the other $5,000$ are maintained to be not paraphrases. We also use the adversarial test sets $PAWS_{QQP}$ and $PAWS_{WIKI}$ form Zhang et al. [97].

**SNLI:** SNLI [6] is a natural language inference (NLI) task. It is a 3-way classification task to classify the inferential relation between two given sequences. We use the same train, development, and IID test set splits as in Chowdhury and Caragea [10]. Any data with a sequence of length $\geq 150$ is filtered out from the training set for efficiency. We use also additional test set splits for stress tests. We use the hard test set split from Gururangan et al. [28], the break test set from Glockner et al. [22], and the counterfactual test set from Kaushik et al. [38].

**MNLI:** MNLI [89] is another NLI dataset, which is similar to SNLI in format. We use the original development sets (match and mismatch) as test sets. We filter out all data with any sequence length $\geq 150$ from the training set. Our actual development set is a random sample of $10,000$ data-points from the filtered training set. As additional testing sets, we use the development set of Conjunctive NLI (ConjNLI) [67] and a few of the stress sets from Naik et al. [56]. These stress test sets include - Negation Match (NegM), Negation Mismatch (NegMM), Length Match (LenM), and Length Mismatch (LenMM). NegM and NegMM add tautologies containing "not" terms - this can bias the models to classify contradiction as the inferential relation because the training set contains spurious correlations between existence of "not" related terms and the class of contradiction. LenM and LenMM add tautologies to artificially increase the lengths of the samples without changing the inferential relation class.

---

[6]https://data.quora.com/First-Quora-Dataset-Release-QuestionPairs

Table 4: Mean accuracy and standard deviation on the Logical Inference [7] for $\geq 10$ number of operations after training on samples with $\leq 6$ operations, and on SST5 [79] and IMDB [49]. **Count.** represents counterfactual test split from Kaushik et al. [38] and **Cont.** represents contrast test split from Gardner et al. [21] The best results are shown in bold. Our models were run 3 times on different seeds. Subscript represents standard deviation. As an example, $90_1 = 90 \pm 0.1$

| Model | Logical Inference Number of Operations | | | SST5 | IMDB | | |
|---|---|---|---|---|---|---|---|
| | 10 | 11 | 12 | IID | IID | Cont. | Count. |
| GT-GRC | $90.33_{22}$ | $88.43_{18}$ | $85.70_{24}$ | $51.67_{8.8}$ | $85.11_{1}0$ | $70.63_{21}$ | $81.97_{5}$ |
| EGT-GRC | $75.79_{61}$ | $73.38_{68}$ | $69.68_{7.8}$ | $51.63_{14}$ | $86.58_{2.7}$ | $72_{9.2}$ | $81.76_{14}$ |
| CRvNN | $94.51_{2.9}$ | $\mathbf{94.48_{5.6}}$ | $92.73_{15}$ | $51.75_{11}$ | $91.47_{1.2}$ | $\mathbf{77.80_{15}}$ | $\mathbf{85.38_{3.5}}$ |
| OM | $94.95_{2}$ | $93.9_{2.2}$ | $93.36_{6.2}$ | $52.30_{2.7}$ | $\mathbf{91.69_{0.5}}$ | $76.98_{5.8}$ | $83.68_{7.8}$ |
| BT-GRC | $95.04_{2.3}$ | $94.29_{3.8}$ | $93.36_{2.4}$ | $\mathbf{52.32_{4.7}}$ | $91.29_{1.2}$ | $75.07_{29}$ | $82.86_{23}$ |
| BT-GRC OS | $\mathbf{95.43_{4.5}}$ | $94.21_{6.6}$ | $\mathbf{93.39_{1.5}}$ | $51.92_{7.2}$ | $90.86_{9.3}$ | $75.68_{21}$ | $84.77_{11}$ |
| EBT-GRC | $94.95_{1.5}$ | $93.87_{7.4}$ | $93.04_{6.7}$ | $52.22_{1}$ | $91.47_{1.2}$ | $76.16_{17}$ | $84.29_{12}$ |

## C   Additional Results

In Table 4, we show that our EBT-GRC model can keep up fairly well with BT-GRC and BT-GRC OS on logical inference [7] and sentiment classification tasks like SST5 [79], and IMDB [21] while being much more computationally efficient as demonstrated in the main paper. Additional comparisons with other models like Transformers and Universal Transformer in logical inference can be found in prior works Shen et al. [71], Tran et al. [84]. They underperform RNNs and RvNNs in logical inference.

## D   Extended Related Works

**RvNN History:** Recursive Neural Networks (RvNNs) in the more specified sense of building representations through trees and directed acyclic graphs were proposed in [63, 24]. Socher et al. [77, 78, 79] extended the use of RvNNs in Natural Language Processing (NLP) by considering constituency trees and dependency trees. A few works [98, 80, 43, 99] started adapting Long Shot-term Memory Networks [33] as a cell function for recursive processing. Le and Zuidema [44], Maillard et al. [51] proposed a chart-based method for simulating bottom-up Recursive Neural Networks through dynamic programming. Shi et al. [75], Munkhdalai and Yu [55] explored heuristics-based tree-structured RvNNs.

RvNNs can also be simulated by stack-augmented recurrent neural networks (RNNs) to an extent (similar to how pushdown automata can model context-free grammar [69, 40]). There are multiple works on stack-augmented RNNs [8, 93, 50]. Ordered Memory [71] is one of the more modern such examples. More recently, DuSell and Chiang [17, 18] explored non-deterministic stack augmented RNNs and Del'etang et al. [14] explored other expressive models. Wu [90] presented a survey of latent structure models.

Choi et al. [9] proposed a greedy search strategy based on easy-first algorithm [23, 48] for auto-parsing structures for recursion utilizing STE gumbel softmax for gradient signals. Peng et al. [61] extended the framework with SPIGOT and Havrylov et al. [31] extended it with reinforcement learning (RL). Ray Chowdhury and Caragea [66] extended it with beam search and soft top-k. Chowdhury and Caragea [10], Zhang et al. [95] introduced different forms of soft-recursion.

**Top-down Signal:** Similar to us, Teng and Zhang [82] explored bidirectional signal propagation (bottom-up and top-down). However, they sent top-down signal in a sequential manner which can be expensive - either it can get slow without parallelization or memory-wise expensive with parallelization of contextualization of nodes in the same height. Our approach in EBT-GAU also has some kinship with BP-Transformer [92]. BP-Transformer allows message passing between a fixed subset of parent nodes and terminal nodes created using a heuristics-based balanced binary tree. Chart-based models can also create sequence contextualized representations [15, 16] but they can be quite expensive by default [66] needing their own separate techniques [34, 35].

**Algorithm 1** Efficient Beam Tree Cell (without slicing)

---

**Input:** data $X = [x_1, x_2, ....x_n]$, $k$ (beam size)
$BeamX \leftarrow [X]$
$BeamScores \leftarrow [0]$
**while** True **do**
  **if** $len(BeamX[0]) == 1$ **then**
    $BeamX \leftarrow [beam[0]$ for $beam$ in $BeamX]$
    break
  **end if**
  **if** $len(BeamX[0]) == 2$ **then**
    $BeamX \leftarrow [cell(beam[0], beam[1])$ for $beam$ in $BeamX]$
    break
  **end if**
  $NewBeamX \leftarrow []$
  $NewBeamScores \leftarrow []$

  **for** $Beam, BeamScore$ in $zip(BeamX, BeamScores)$ **do**
    $Scores \leftarrow log \circ softmax([scorer(beam[i], beam[i+1])$ for $parent$ in $Parents])$
    $Indices \leftarrow topk(Scores, k)$

    **for** $i$ in $range(K)$ **do**
      $newBeam \leftarrow deepcopy(Beam)$
      $newBeam[Indices[i]] \leftarrow cell(Beam[Indices[i]], Beam[Indices[i]+1])$
      Delete $newBeam[Indices[i]+1]$
      $NewBeamX.append(newBeam)$
      $newScore \leftarrow BeamScore + Scores[indices[i]]$
      $newBeamScores.append(newScore)$
    **end for**
  **end for**
  $Indices \leftarrow topk(newBeamScores, k)$
  $BeamScores \leftarrow [newBeamScores[i]$ for $i$ in Indices$]$
  $BeamX \leftarrow [newBeamX[i]$ for $i$ in Indices$]$
**end while**
$BeamScores \leftarrow Softmax(BeamScores)$
Return $sum([score * X$ for $score, X$ in $zip(BeanScores, BeamX)])$

---

**Transformers + RvNNs:** There have been several approaches to incorporating RvNN-like inductive biases to Transformers. For instance, Universal Transformer [13] introduced weight-sharing and dynamic halt to Transformers. Csordás et al. [12] extended on universal transformer with geometric attention for locality bias and gating. Shen et al. [74] built on weight-shared transformers with high layer depth and group self-attention. Wang et al. [87], Nguyen et al. [58], Shen et al. [73] added hierarchical structural biases to self-attention. Fei et al. [20] biased pre-trained Transformers to have constituent information in intermediate representations. Hu et al. [34] used Transformer as binary recursive cells in chart-based encoders.

# E   Pseudocode

We present the pseudocode of EBT-RvNN in Algorithm 1. Note that the algorithms are written as they are for the sake of illustration: in practice, many of the nested loops are made parallel through batched operations.

# F  Architecture details

## F.1  Sentence Encoder Models

For the sentence encoder models the architectural framework we use is the same siamese dual-encoder setup as Ray Chowdhury and Caragea [66].

## F.2  Sentence Interaction Models

**GAU-Block:** Our specific implementation of a GAU-block [36] is detailed below. Our GAU-Block can be defined as $\text{GAUBlock}(x, p, G)$. The function arguments are of the following forms: $x \in \mathbb{R}^{n \times d}, p \in \mathbb{R}^{l \times d}$ and $G \in \{0, 1\}^{n \times l}$. $x$ accepts the main sequence of vectors that is to serve as attention queries; $p$ accepts either the sequence of intermediate node representations created from our RvNN (for parent attention) or it accepts the same input as $x$ (for usual cases); $p$ serves as keys and values for attention; $G$ accepts either the adjacency matrix in case of parent attention (where $G_{ij} = 1$ iff $p_j$ is a parent of $x_i$ else $G_{ij} = 0$), otherwise, it accepts just the usual attention mask; either way, $G$ serves as an attention mask.

$$x' = LN(xW_{init} + b_{init}); \;\; p' = LN(pW_{init} + b_{init}) \tag{8}$$

$$u = \text{SiLU}(x'W_u + b_u); \;\; v = \text{SiLU}(p'W_v + b_v) \tag{9}$$

$$q = z_q \odot \text{SiLU}(x'W_z + b_z) + zb_q; \;\; k = z_k \odot \text{SiLU}(p'W_z + b_z) + zb_k \tag{10}$$

$$A = \text{Softmax}(\frac{qk^T + pos}{\sqrt{2d}}, mask = G) \tag{11}$$

$$v' = Av \tag{12}$$

$$o = (u \odot v')W_o + b_o \tag{13}$$

$$g = \text{Sigmoid}([o; x]W_{gate} + b_{gate}) \tag{14}$$

$$out = g \odot o + (1 - g) \odot x \tag{15}$$

Here, $W_{init} \in \mathbb{R}^{d \times d}; W_z \in \mathbb{R}^{d \times d_h}, W_u, W_v \in \mathbb{R}^{d \times 2d}, b_{init}, b_z, b_o \in \mathbb{R}^d; z_q, zb_q, z_k, zb_k \in \mathbb{R}^{d_h}; b_u, b_v \in \mathbb{R}^{2d}, W_o, W_{gate} \in \mathbb{R}^{2d \times d}$. $[;]$ represents concatenation.

$LN$ is layer normalization. $pos$ is calculated using the technique of Raffel et al. [64] using relative tree height distance for parent attention, or relative positional distance for usual cases.

**GAU Sequence Interaction Setup**: Let GAUStack represent some arbitrary number of compositions of GAUBlocks (multilayered GAU block). GAUStack has the same function arguments as GAUBlock. Given two sequences $(x_1, x_2)$ and their corresponding attention masks $(M_1, M_2)$ as inputs where $x_1 \in \mathbb{R}^{n_1 \times d}, x_2 \in \mathbb{R}^{n_2 \times d}, M_1 \in \{0, 1\}^{n_1 \times n_1}, M_1 \in \{0, 1\}^{n_2 \times n_2}$, the GAU setup can be expressed as:

$$inp = [CLS + seg_1; x_1 + seg_1; SEP; CLS + seg_2, x_2 + seg_2] \tag{16}$$

$$r = \text{GAUStack}(x = inp, p = inp, G = f(M_1; M_2)) \tag{17}$$

$$\alpha = \text{Softmax}(\text{GELU}(rW_1 + b_1)W_2 + b_2) \tag{18}$$

$$cls' = \sum_i \alpha_i r \tag{19}$$

$$logits = \text{GELU}(cls'W_1^{logits} + b_1^{logits})W_2^{logits} + b_2^{logits} \tag{20}$$

Here, $CLS, SEP, seg_1, seg_2 \in \mathbb{R}^{1 \times d}$ are randomly initialized trainable vectors; $seg_1, seg_2$ are segment embeddings. $W_1 \in \mathbb{R}^{d \times d}, W_2 \in \mathbb{R}^{d \times 1}; b_1, b_2, b_1^{logits} \in \mathbb{R}^d; b_2^{logits} \in \mathbb{R}^c; W_1^{logits} \in \mathbb{R}^{d \times d}, W_2^{logits} \in \mathbb{R}^{d \times c}$. $c$ is the number of classes for the task. $f$ is a function that takes the attention masks as input and concatenates them while adjusting for the special tokens (CLS, SEP).

**EGT-GAU Sequence Interaction Setup:** EGT-GAU starts from the same input as above. Let us also assume we have the EGT-GRC$(x)$ module which takes a sequence of vectors $x \in \mathbb{R}^{n \times d}$ as the input to recursively process and outputs $(cls, p, G)$ where $cls \in \mathbb{R}^{1 \times d}$ is the root representation, $p \in \mathbb{R}^{l \times d}$ is the sequence of non-terminal representations from the tree, and $G \in \{0, 1\}^{n \times l}$ is the

adjacency matrix for parent attention (i.e., $G_{ij} = 1$ iff $p_j$ is a parent of $x_i$, else $G_{ij} = 0$). Technically, tree height information is also extracted for relative position but we do not express that explicitly for the sake of brevity. With these elements, EGT-GAU can be expressed as below:

$$cls_1, p_1, G_1 = \text{EGT-GRC}(x = x_1); \quad cls_2, p_2, G_2 = \text{EGT-GRC}(x = x_2) \tag{21}$$

$$x'_1 = \text{GAUStack}_1(x = x_1, p = p_1, G = G_1); \quad x'_2 = \text{GAUStack}_1(x = x_2, p = p_2, G = G_2) \tag{22}$$

$$cls'_1 = \text{GELU}(cls_1 W_1^{cls} + b_1^{cls}) W_2^{cls} + b_2^{cls}; \quad cls'_2 = \text{GELU}(cls_2 W_1^{cls} + b_1^{cls}) W_2^{cls} + b_2^{cls} \tag{23}$$

$$inp = [cls'_1 + seg_1; x'_1 + seg_1; SEP; cls'_2 + seg_2, x'_2 + seg_2] \tag{24}$$

$$r = \text{GAUStack}_2(x = inp, p = inp, G = f(M_1, M_2)) \tag{25}$$

Everything else after eqn. 25 is the same as eqn. 18 to 20. $SEP, seg_1, seg_2 \in \mathbb{R}^{1 \times d}$; $seg_1, seg_2$ are segment embeddings as before. $W_1^{cls}, W_2^{cls} \in \mathbb{R}^{d \times d}$; $b_1^{cls}, b_2^{cls} \in \mathbb{R}^d$.

**EBT-GAU Sequence Interaction Setup:** This setup is similar to that of EGT-GAU but with a few changes. EBT-GAU uses EBT-GRC as a module instead of EGT-GRC. EBT-GAU returns outputs of the form $(cls, bp, bG, s)$ where $cls \in \mathbb{R}^{1 \times d}$ is the beam-score-weighted-averaged root representation, $bp \in \mathbb{R}^{b \times l \times d}$ are the beams (beam size $b$) of sequences of non-terminal representations from the tree, $bG \in \{0, 1\}^{b \times n \times l}$ are the beams of adjacency matrices for parent attention, and $s \in \mathbb{R}^b$ are the softmax-normalized beam scores. Let NGAUStack represent the same function as GAUStack but formalized for batched processing of multiple beams of sequences. With these elements, EBT-GAU can be expressed as:

$$cls_1, bp_1, bG_1, s_1 = \text{EBT-GRC}(x = x_1); \quad cls_2, bp_2, bG_2, s_2 = \text{EBT-GRC}(x = x_2) \tag{26}$$

$$bx_1 = \text{repeat}(x_1, b); \quad bx_2 = \text{repeat}(x_2, b) \tag{27}$$

$$bx'_1 = \text{NGAUStack}_1(bx_1, bp_1, bG_1); \quad bx'_2 = \text{NGAUStack}_1(bx_2, bp_2, bG_2) \tag{28}$$

$$x'_1 = \sum_i s[i] \cdot bx'_1[i]; \quad x'_2 = \sum_i s[i] \cdot bx'_2[i] \tag{29}$$

Everything else after eqn. 29 is the same as the equations 23-25 followed by the equations 18 to 20. $\text{repeat}(x, b)$ changes $x \in \mathbb{R}^{n \times d}$ to $bx \in \mathbb{R}^{b \times n \times d}$ by batching the same $x$ for $b$ times.

# G  Hyperparameter details

For sentence encoder models, we use the same hyperparameters as [66] (the preprint of the paper is available in the supplementary in anonymized form) for all the datasets. The only new hyperparameter for EBT-GRC is $d_s$ which we set as 64; otherwise the hyperparameters are the same as that of BT-GRC or BT-GRC OS. We discuss the hyperparameters of the sequence interaction models next. For EBT-GAU/EGT-GAU, we used a two-layered weight-shared GAU-Blocks for NGAUStack$_1$/GAUStack$_1$ and a three-layered weight-shared GAU-Blocks for GAUStack$_2$ (for parameter efficiency and regularization). GAU uses a five-layered GAU-Blocks (weights unshared) for GAUStack so that the parameters are similar to that of EBT-GAU or EGT-GAU. We use a dropout of 0.1 after the multiplation with $W_o$ in each GAUBlock layer and a head size $d_h$ of 128 (similar to Hua et al. [36]). For relative position, we set $k = 5$ ($k$ here corresponds the receptive field for relative attention in Shaw et al. [70]) for normal GAUBlocks and $k = 10$ for parent attention (since parent attention is only applied to higher heights, we do not need to initialize weights for negative relative distances). Other hyperparameters are kept same as the sentence encoder models. The hyperparameters of MNLI, SNLI, and QQP are shared. Note that all the natural language tasks are trained with fixed 840B Glove Embeddings [62] as in Ray Chowdhury and Caragea [66]. All models were trained in a single Nvidia RTX A6000. The code is available in the supplementary.

