# OpenReview forum: "Efficient Beam Tree Recursion"
_NeurIPS.cc/2023/Conference — NeurIPS 2023 poster_

### Official Review · Reviewer_S18L · 2023-07-06

**Soundness:** 4 excellent
**Presentation:** 3 good
**Contribution:** 3 good
**Rating:** 5
**Confidence:** 3

**Summary:**

This paper relates to tree-recursive neural networks, which are a type of network where network modules are wired up in a tree structure that is itself determined by outputs of earlier parts of the network. This recursion poses significant challenges, including speed, memory usage, and potentially other issues (such as the need to use reinforcement learning to train the selection of tree structure). This paper proposes a more lightweight architecture for the portion of the network that determines the topology of the tree, in the context of a family of approaches that have good behavior in terms of gradient estimation. As a result, the method achieves both good memory usage and time complexity.

Additionally, the paper proposes the use of tree-recursive networks to produce representations for each token, instead of a single representation that summarizes the entire input.

**Strengths:**

The proposed method performs well in terms of speed, memory usage, and out-of-distribution generalization on ListOps -- it seems to be the overall winner compared to the baseline recursive NN methods.

**Weaknesses:**

One concern regarding Tree-RvNNs is that the efficiency limitations set put this entire class of methods outside of practical usefulness for many tasks. The paper is notably light in comparisons to other families of approaches, such as the standard Transformer. This weakness is most pronounced in the part of the paper dealing with token representations and evaluating on SNLI/QQP/MNLI, which are standard tasks to run Transformers on (unlike ListOps, where the applicability of Tree-RvNNs is readily apparent).

The part of the paper that deals with token-level encoding (Section 4) seems underdeveloped, at least compared to the discussion of Tree-RvNNs in the sentence-encoding scenarios. As a result the paper gives a bit of an impression of two distinct ideas ideas being glued together, as opposed to a more coherent direction for the work.

**Questions:**

How does the proposed approach compare to more standard architectures like the Transformer?

**Limitations:**

No concerns regarding discussion of limitations.

---

> ### Author Rebuttal · Authors · 2023-08-09
>
> Thank you for the review.
>
> **Transformer Comparison:**
>
> In Logical Inference (Results in Table 1 Appendix), Transformers were shown to perform poorly in the same length generalization settings.  Tran et al. [1] showed that the performance of Transformers falls sharply to around 40% when generalizing to 12 logical operators settings. Shen et al. [2] managed to achieve around 48% with Transformers and Universal Transformers in similar settings. On the other hand, our proposed methods maintain 93%+.
>
> Shen et al. [2] also showed Universal Transformers get only about 71.5% in the IID (let alone OOD) settings in ListOps; and Transformers perform even worse (~57%). In the appendix paper “Beam Tree Recursive Cells”, we also show that a more advanced Transformer model with RvNN-like biases (Neural Data Router [3]) is still deficient in more extreme OOD generalization even after 10 times more training data (Table 7). In contrast, our method is near perfect in length generalization.
>
> Comparison in realistic data is more challenging in a parameter-controlled manner (increasing the parameters of RvNN is difficult without the section 4 approach for multi-layering otherwise increasing the parameters of recursive cells can make it very slow. On the other hand, reducing the parameters of Transformers can unnecessarily handicap it. Moreover, Transformers shine the best on those tasks with pre-training - which is a regime yet to be explored with RvNN). Note that this limitation (of lack of comparison of Transformers in realistic data) is also shared in other works on modern RvNNs [2,5] or RvNN-like Transformers [3].
>
> Nevertheless, for the token representation setting that you were asking for we do show, in Table 3, EBT-GAU (Our method + Transformer hybrid) vs GAU (Pure Transformer) in parameter-controlled settings. GAU is a popular modern Transformer variant [4] that is becoming a new standard (for example, it was adopted in the recent MEGA [6] as well).
>
> [1] The Importance of Being Recurrent for Modeling Hierarchical Structure - Tran et al. EMNLP 2018
>
> [2] Ordered Memory - Shen et al. Neurips 2019
>
> [3] The Neural Data Router: Adaptive Control Flow in Transformers Improves Systematic Generalization - Csordas et al. ICLR 2022
>
> [4] Transformer Quality in Linear Time - Hua et al. ICML 2022
>
> [5] Modeling Hierarchical Structures with Continuous Recursive Neural Networks - Ray Chowdhury et al. ICML 2021
>
> [6] Mega: Moving Average Equipped Gated Attention - Ma et al. ICLR 2023
>
> **Section 4:**
>
> We will improve the presentation of section 4. Appendix 5.2 develops the section with more details.
>
> **Practical Use:**
>
> RvNNs can be used for practical sentence encoding tasks just as anything else. It will be slower (not unusable though) - but even in that direction, they can be improved. In another of our upcoming works, we managed to make EBT-RvNN significantly faster.

---

> > ### Comment · Reviewer_S18L · 2023-08-19
> >
> > Thank you for the response.
> >
> > The additional information in the response helps put the results in context. I still think that the lack of discussion of Transformer baselines in the paper itself can make it harder to contextualize for a reader who is not already familiar with the line of work on recursive networks and related architectures.

---

> > > ### Author Response · Authors · 2023-08-19
> > >
> > > Thank you for the feedback. We will add the additional results for better contextualization.

---

### Official Review · Reviewer_XtZN · 2023-07-09

**Soundness:** 3 good
**Presentation:** 2 fair
**Contribution:** 3 good
**Rating:** 7
**Confidence:** 3

**Summary:**

This paper makes 2 contributions. The major one is a more efficient Beam Tree Recursive Neural Network (BT-RvNN) that learns a binary tree structure over a sequence of tokens. The second is a cross attention mechanism to incorporate this tree structure into transformers.

They increase the efficiency by decreasing the hidden dimension and increasing the parallelizability and remove OneSoft. They verify that they do indeed see memory usage drop and can apply their method to longer sequences.

They achieve minimal or no loss in quality with their new efficient method when using it as a sentence encoder. Their model seems to work especially well in the sentence interaction context when they preserve the token representation.

**Strengths:**

Good exposition and necessary background provided.

Efficiency results are very strong and the method clearly achieves a convincing win with memory usage.

Experiments are well done a reproducible.

**Weaknesses:**

The part about sentence interactions detracted from the clarity of the paper since I think the main contribution is the efficiency wins.

The method on occasion does perform worse?

**Questions:**

I don't complete understand section 4. Is the model jointly trained with the transformer or is the model trained separately and frozen and the tree structure is what is consumed?

When using EBT-GAU vs EGT-GAU in the sentence interaction models, the difference is in the hidden states h_{i]^{t} of the tree nodes?

**Limitations:**

Method is still slow.

---

> ### Author Rebuttal · Authors · 2023-08-09
>
> Thank you for the review.
>
> Re Weaknesses:
>
> 1. We will improve Section 4, but we would like to point out that we already present more details in Appendix 5.2 [*] about the ideas from Section 4. If there are further questions, we will be happy to answer them here.
>
> 2. The target of the approach is to reduce memory usage significantly while approximately preserving the accuracy performance against the original “non-efficient” approaches. As such, our approach can occasionally perform slightly better or slightly worse given that we have not introduced anything to guarantee better accuracy explicitly compared to the non-efficient variants.
>
> Re Questions:
>
> 1. Yes, the model is jointly trained (end-to-end) with the transformer (not trained separately and no freezing is done).  Appendix 5.2 expands on section 4 more.
>
> 2. Yes, in the sense that EGT-GAU tree nodes are constructed by Greedy search, whereas EBT-GAU tree nodes are constructed by Beam Search. Another difference is that the former creates nodes of a single tree (L126-130 Appendix), and the latter (L139-143 Appendix) creates separate node representations per tree structure in each beam - which are later marginalized based on beam scores (eqn. 22 in Appendix).
>
> [*] There is a minor error in the Appendix. We set [M1;M2] as the input mask for the sequence concatenations in Appendix in Eqn. 10,18 but this misses the mask values for CLS and SEP special tokens. We will fix that in the final version.

---

### Official Review · Reviewer_oTCB · 2023-07-21

**Soundness:** 2 fair
**Presentation:** 3 good
**Contribution:** 2 fair
**Rating:** 5
**Confidence:** 3

**Summary:**

This paper presents a comprehensive analysis of Beam Tree Recursive Neural Networks (BT-RvNN), a recently proposed variant of recursive neural networks. The study identifies suboptimal neural network parameterization in BT-RvNNs, which adversely affects running efficiency and results in excessive memory consumption. By proposing several techniques to remove the memory bottleneck, the authors demonstrate that BT-RvNNs can be run with significantly reduced memory while still maintaining task performance. Additionally, the paper presents an attention-augmented approach to expand the score of Tree-RvNNs beyond sentence encoders to encompass sentence contextualization, enriching the potential downstream applications of Tree-RvNNs.

**Strengths:**

- The paper is well structured and offers an extensive background of recursive neural networks.
- The proposed generalization from sentence encoders to sentence contextualizers is compelling, showcasing promising empirical performance compared to plain attention modules without tree-RvNNs. This expansion broadens the potential applications of Tree-RvNNs and establishes them as building blocks with specific inductive biases for learning tasks.

**Weaknesses:**

- My main concern lies in the clarity of motivation. If my understanding is correct, the strategies proposed to enhance efficiency seem to be primarily related to improving neural network parameterization, rather than improving the beam tree recursion algorithm, which may have been inferred during the reading of the introduction. As a result, the motivation could be perceived more as an engineering effort with limited technical novelty.
- Although the proposed method demonstrates a significant reduction in memory usage, the trade-off between runtime and memory utilization becomes evident in both baselines (GT-GRC and BT-GRC). A more in-depth exploration of this trade-off and its implications would enhance the comprehensiveness.

**Questions:**

1. Why does the simplified network parameterization outperform the original architecture? An in-depth examination of this (e.g., conducting ablation studies of proposed fixes) would contribute to a more comprehensive understanding of the underlying mechanisms.
2. The experimental findings reveal that EGT-GRC exhibits a considerable performance drop compared to the original GT-GRC across diverse tasks, whereas EBT-GRC outperforms BT-GRC on most tasks. Why is this the case? The paper would benefit from providing a thorough discussion of the factors influencing the contrasting performance trends observed in EGT-GRC and EBT-GRC.
3. Typo:
    L269: missing close parenthesis.

**Limitations:**

The authors have adequately addressed the limitations.

---

> ### Author Rebuttal · Authors · 2023-08-09
>
> Thank you for the review.
>
> Regarding Weaknesses:
>
> 1. Both BT-RvNN and Gumbel Tree RvNN [1] are rooted in an underlying easy-first parsing algorithm utilized in [1] (BT-RvNN is an extension of it using top-k operators and pruning). We improve the algorithm as used in [1] which also allows improvement of BT-RvNN. It’s not an improvement of parameter count but an improvement of how computation is allocated. The main improvement is the removal of the need for parallel application of the recursive cell function for all positions for scoring in every recursive iteration. Our modification allows employing the heavy recursive cell function to be applied only at one position per iteration - resulting in significant memory utilization reduction. The additional improvement from slicing is a bonus. We will improve the clarity of the motivation in the paper.
>
> 2. GT-GRC/BT-GRC are only slightly faster than EGT-GRC/EBT-GRC. There isn’t a deep reason for that other than just differences in overhead - for example, EGT-GRC/EBT-GRC have to run an MLP scorer and then run the recursive cell, whereas GT-GRC/BT-GRC just run the recursive cell + single linear transformation on top of it for both scoring and the updated representation (this saves an extra layer of transformation per iteration).
>
> Regarding Questions:
>
> 1. It’s possible that the disentanglement of the scorer function and the recursive cell as discussed in the paper enables a better division of labor for the parameters for ListOps. Moreover, slicing can provide an extra boost possibly by inducing an information bottleneck effect by limiting the dimensions of scorer inputs (and as we discussed in the paper, the scoring decision would hinge on more abstract type-level features which may be better represented with lower dimensions for mitigating overfitting). Ultimately, our main target was simply to reduce memory cost - the accuracy improvement in ListOps is more of a serendipity.  And the improvement is not consistent across the tasks - the big improvement is mainly in ListOps.
>
> We also ran an additional ablation with no slicing here as per your requests (on ListOps):
>
> | Models            | IID    | 200- 300 | 500-600 | 900-1000 | 10    | 15    | LRA   |
> |-------------------|--------|----------|---------|----------|-------|-------|-------|
> | BT-GRC            | 99.39  | 96.15    | 92.55   | 86.7     | 77.1  | 63.7  | 67.3  |
> | EBT-GRC w/o slice | 99.87  | 99.35    | 99.5    | 99.6     | 79.25 | 74.35 | 75.35 |
> | EBT-GRC           | 99.9   | 99.85    | 99.4    | 99.5     | 82.95 | 79    | 79.5  |
>
> 2. There is no consistent observation that EGT-GRC always underperforms GT-GRC. EGT-GRC does better than GT-GRC in some sections of ListOps, e.g. IID and length generalization 200-300. It also does well in semantic classification that we show in the Appendix (Table 1, SST5, IMDB). Overall, both the gumbel tree models, in structure-sensitive contexts, fail to model the task invariances properly leading to failure in length generalization. It’s hard to tell beyond that why one model fails “more” than the other in certain contexts. The non-efficient GT-GRC, however, does allow the use of the full hidden state features to store information relevant for scoring in a more direct manner - which is perhaps beneficial in some contexts when the structural signal is poorer (due to lack of beam search).
>
> [1]  Learning to Compose Task-Specific Tree Structures - Choi et al. AAAI 2018

---

> > ### Comment · Reviewer_oTCB · 2023-08-16
> >
> > I thank the authors for the feedback. The provided clarification and new experimental results make the context clearer to me and resolve some of my concerns (especially Weakness 2 and Questions 1 and 2). However, I am still not fully convinced by the motivation. While it is true that the memory efficiency is largely improved, it appears that the advancements primarily come from an improved neural network parameterization (it is not about parameter count but a more suitable parameter allocation). In this regard, the technical contribution seems limited, as it focuses more on designing a neural network that implements the algorithm with less memory, rather than improving the algorithm itself.
> >
> > After carefully reading through the clarification as well as the other reviews, I decided to stay at my initial rating of 5.

---

> > > ### Author Response · Authors · 2023-08-17
> > > **Clarifications**
> > >
> > > Thank you for the response.
> > > Based on the response we would like to be clear and explicit about a few points:
> > >
> > > 1. **On Parameterization:** You are right that the EBT-RvNN and EGT-RvNN are, in a sense, better parameterization (better allocation of compute) of BT-RvNN and GT-RvNN, respectively. However, contributions in similar spaces appear in  NeurIPS [1].
> > >
> > > 2. **On the clarity of the motivation:** The motivation for EBT-RvNN simply is memory reduction (as you noted can be inferred from the abstract) while preserving accuracy. Our specific form of parameterization is proposed and demonstrated as an effective means to achieve the motivated goal.
> > >
> > > 3. **On change in the algorithm (Part 1):** Whether the improvement (improvement in memory reduction is also an improvement) is made on the side of the algorithm or elsewhere depends on how abstract the algorithm we have in mind. This can be a subjective point. Precisely, it depends on whether we want to consider the use of next parent representations as input to the score function in Choi et al. as a mere "implementation detail" or as part of the specific algorithm used in Choi et al.  We would admit, however, that we are not changing the more general idea of easy-first parsing algorithm.
> > >
> > > 4. **On change in the algorithm (Part 2):** If we fix the level of detail to be that in which Algorithm 1 (Greedy Search-based RvNN) and Algorithm 2 (Beam Search based RvNN) are presented in Appendix A of the “Beam Tree Recursive Cell” paper (from supplementary material) [2], then technically the parameterization does involve changes of several lines in the algorithms. For example, the line ```Parents← [cell(childL, childR) for childL, childR in zip(ChildrenL,ChildrenR]``` will be entirely removed, and there will be separate lines for selecting the chosen sibling pairs and apply the cell function to them alone besides changing the scorer. We will add an updated pseudocode for better contrast against Algorithm 1 and 2 as presented in “Beam Tree Recursive Cell”.
> > >
> > > 5. **Other contributions:** We also want to highlight that, even if the technical contribution of EBT-RvNN may by itself be considered limited, this is not the sole contribution. As you noted (in strengths), we also introduce the parent attention mechanism for creating token contextualized representations which further adds to our overall technical contribution (EBT-RvNN + parent attention). This is not just a change of parameterization but requires further non-trivial technical changes (Appendix 5.2) and additionally paves the way for broader applicability of EBT-RvNN (as you acknowledged).
> > >
> > > [1] FlashAttention: Fast and Memory-Efficient Exact Attention with IO-Awareness - Dao et al. NeurIPS 2022
> > >
> > > [2] Anonymized paper; published in ICML 2023

---

### Official Review · Reviewer_x8Vp · 2023-07-26

**Soundness:** 3 good
**Presentation:** 3 good
**Contribution:** 2 fair
**Rating:** 5
**Confidence:** 4

**Summary:**

1. This paper identify a critical memory bottleneck in both Gumbel-Tree RvNN and Beam-Tree RvNN and propose effective strategies to address it. The approaches reduce the peak memory usage of Beam-Tree RvNNs by 10-16 times in certain stress tests (refer to Table 4).
2. This paper introduce a novel strategy that utilizes intermediate tree nodes (span representations) to provide top-down signals to the original terminal representations using a parent attention mechanism.
3. This paper demonstrate that the proposed efficient variant of BT-RvNN incurs minimal accuracy loss compared to the original and, in some cases, even outperforms the original by a significant margin (especially in ListOps).

**Strengths:**

1. This paper identify a critical memory bottleneck in both Gumbel-Tree RvNN and Beam-Tree RvNN and propose effective strategies to address it. The approaches reduce the peak memory usage of Beam-Tree RvNNs by 10-16 times in certain stress tests (refer to Table 4).
2. This paper introduce a novel strategy that utilizes intermediate tree nodes (span representations) to provide top-down signals to the original terminal representations using a parent attention mechanism.
3. This paper demonstrate that the proposed efficient variant of BT-RvNN incurs minimal accuracy loss compared to the original and, in some cases, even outperforms the original by a significant margin (especially in ListOps).

**Weaknesses:**

The paper is written very clearly, and the methods are simple yet effective.

**Questions:**

1. If some baselines based on sequence-to-sequence methods are provided, the motivation for the Tree-based model will become clearer.

**Limitations:**

Yes.

---

> ### Author Rebuttal · Authors · 2023-08-09
>
> Thank you for the review.
>
> We are currently focusing on encoder-focused tasks (classification, logical inference) like prior works on modern RvNNs [1,2,3] for which there is no seq2seq baseline (unless we use pre-trained models + prompting but they would not be fair comparisons). We agree that it is an interesting question as to how well our method can be adapted for seq2seq processing which we leave open for future exploration. This would require more extensive analysis (best suited for a separate paper) because of the lack of precedence in the application and comparison of RvNN-based models in seq2seq in a straightforward standardized manner.
>
> [1] Ordered Memory - Shen et al. Neurips 2019
>
> [2] Modeling Hierarchical Structures with Continuous Recursive Neural Networks - Ray Chowdhury et al. ICML 2021
>
> [3] Learning to Compose Task-Specific Tree Structures - Choi et al. AAAI 2018

---

### Decision · Program_Chairs · 2023-09-21

**Decision:**

Accept (poster)

**Comment:**

This paper addresses the memory usage challenges in the Beam Tree Recursive Neural Network (BT-RvNN) and identifies the main bottleneck as the entanglement of the scorer function and the recursive cell function. The reviewers agree that the paper is well-written and making a solid contribution to the community. I therefore would recommend accept this paper.